# Chronic Exposure to High Fat Diet Affects the Synaptic Transmission That Regulates the Dopamine Release in the Nucleus Accumbens of Adolescent Male Rats

**DOI:** 10.3390/ijms24054703

**Published:** 2023-02-28

**Authors:** Wladimir Plaza-Briceño, Victoria B. Velásquez, Francisco Silva-Olivares, Karina Ceballo, Ricardo Céspedes, Gonzalo Jorquera, Gonzalo Cruz, Jonathan Martínez-Pinto, Christian Bonansco, Ramón Sotomayor-Zárate

**Affiliations:** 1Centro de Neurobiología y Fisiopatología Integrativa (CENFI), Instituto de Fisiología, Facultad de Ciencias, Universidad de Valparaíso, Valparaíso 2360102, Chile; 2Programa de Doctorado en Ciencias Mención Neurociencias, Facultad de Ciencias, Universidad de Valparaíso, Valparaíso 2360102, Chile; 3Instituto de Ciencias Naturales, Facultad de Medicina Veterinaria y Agronomía, Universidad de las Américas, Viña del Mar 2520000, Chile

**Keywords:** dopamine, accumbens, electrophysiology, HFD, FSCV, neuroinflammation

## Abstract

Obesity is a pandemic caused by many factors, including a chronic excess in hypercaloric and high-palatable food intake. In addition, the global prevalence of obesity has increased in all age categories, such as children, adolescents, and adults. However, at the neurobiological level, how neural circuits regulate the hedonic consumption of food intake and how the reward circuit is modified under hypercaloric diet consumption are still being unraveled. We aimed to determine the molecular and functional changes of dopaminergic and glutamatergic modulation of nucleus accumbens (NAcc) in male rats exposed to chronic consumption of a high-fat diet (HFD). Male Sprague-Dawley rats were fed a chow diet or HFD from postnatal day (PND) 21 to 62, increasing obesity markers. In addition, in HFD rats, the frequency but not amplitude of the spontaneous excitatory postsynaptic current is increased in NAcc medium spiny neurons (MSNs). Moreover, only MSNs expressing dopamine (DA) receptor type 2 (D_2_) increase the amplitude and glutamate release in response to amphetamine, downregulating the indirect pathway. Furthermore, NAcc gene expression of inflammasome components is increased by chronic exposure to HFD. At the neurochemical level, DOPAC content and tonic dopamine (DA) release are reduced in NAcc, while phasic DA release is increased in HFD-fed rats. In conclusion, our model of childhood and adolescent obesity functionally affects the NAcc, a brain nucleus involved in the hedonic control of feeding, which might trigger addictive-like behaviors for obesogenic foods and, through positive feedback, maintain the obese phenotype.

## 1. Introduction

Obesity is a health problem worldwide characterized by an excess of fatty tissue and metabolic alterations [1]. In 2016, the World Health Organization (WHO) stated that globally, more than 1900 million adults (>18 years old) were overweight, and 650 million of these were obese [1]. In addition, 340 million children and adolescents (5 to 19 years old) were overweight or obese, representing an increase in the prevalence of overweight and obesity from 4% in 1975 to more than 18% in 2016 [1]. In Chile, according to The National Health Survey (2016–2017), 74.2% of the population over 15 years old, shows some degree of weight excess, and 34.4% of this population has obesity [2]. In this context, the search for new pharmacological treatments and targets for obesity remains a hot topic, mainly when anorectic agents used to combat obesity have been associated with severe adverse events, withdrawing them from the market (e.g., fenfluramine, dexfenfluramine, and sibutramine) [3]. Among the causes of this pandemic are an unhealthy lifestyle and an excess of obesogenic food intake characterized by hypercaloric content and high palatability [4]. Chronic exposure to this kind of diet affects brain areas such as the hypothalamus (associated with homeostatic eating control) and the reward system (related to hedonic eating control), leaving a predisposition to the development of overweight and obesity [5]. In some cases, the hyperstimulation of the reward system by obesogenic food (rich in fats and carbohydrates) could promote dependence or food addiction [6,7,8].

Dopamine (DA) neurons form the reward system or mesocorticolimbic circuit from the ventral tegmental area (VTA) whose efferents go to limbic and cortical regions such as nucleus accumbens (NAcc) and prefrontal cortex (PFC), respectively [9,10]. The VTA DA neurons are tonically inhibited by GABA interneurons [11,12]. Still, natural rewards (e.g., food, water, sex, and social interaction) and drugs of abuse (e.g., cocaine, amphetamine, ethanol, nicotine, and morphine, among others) disinhibit DA neurons, increase their firing rate, and DA release in NAcc and PFC [9,13,14,15]. In NAcc, the medium spiny neurons (MSNs), which are some kinds of GABAergic inhibitory neurons, are segregated by their electrical and synaptic properties into those that express preferentially DA receptor type 1 (D_1_) or type 2 (D_2_) [16,17]. It has been suggested that both D_1_- and D_2_-expressing MSNs are part of the GABAergic balance that regulates the DA release of DA from VTA [18,19]. In addition, VTA DA neurons are activated by orexinergic/glutamatergic neurons from the lateral hypothalamus (LH) [20], also promoting DA release in the NAcc [21]. Interestingly, LH orexinergic neurons’ activity is strongly controlled by inhibitory neurons from the lateral septum (LS), which also activate the firing of VTA DA neurons by inhibiting VTA GABA interneurons [22,23,24].

Feeding behavior is essential for the species’ survival and is highly regulated by integrating homeostatic and hedonic signals [25]. Chronic stimuli such as obesogenic food (rich in nutrients such as salt, fat, or sweets) promote maladaptive eating behavior that leads to eating disorders or obesity [26]. During the last decade, several works have studied that the reward system and other brain areas are susceptible to inflammation by chronic exposure to stimuli such as drug abuse and hypercaloric diets. For example, chronic exposure to ethanol (2 weeks) increases protein levels of tumor necrosis factor α (TNF-α) and interleukin-17A (IL-17A) in the PFC of female adolescent mice [27]. In contrast, one injection of methamphetamine (10 mg/kg) in rats increases protein levels of interleukin-1β (IL-1β) in VTA, NAcc, and PFC [28]. On the other hand, neuroinflammation in the reward system induced by obesogenic diets has been less studied. Recently, it has been shown that prolonged exposure to a cafeteria diet in male mice for six weeks increases mRNA expression of IL-1β and interferon-γ (IFN-γ) and microglial activation in NAcc [29], while exposure to a high-fat diet (HFD) for 12 weeks produces an increase in body weight and plasma levels of leptin, insulin, glycemia, C-reactive protein, and NAcc gene expression of IKKβ (Inhibitor of Nuclear Factor Kappa B Kinase Subunit Beta, a kinase that activates NF-κB), glial fibrillary acidic protein (GFAP), ionized calcium-binding adapter molecule 1 (Iba-1), IL-1β, IFN-γ, and cluster of differentiation 45 (CD45) [30].

In summary, obesity induces pathophysiological alterations in brain areas associated not only with homeostatic control of food intake but also in nuclei of hedonic control of eating, where patterns of neurotransmitter release, gene expression, and synaptic communications are affected. In this work, we will unravel the molecular, electrophysiological, and neurochemical deregulations in NAcc associated with the childhood and adolescent obesity model. In addition, this work opens the possibility of studying NAcc as a pharmacological target to regulate hedonic food intake.

## 2. Results

### 2.1. Murinometric Parameters in Rats Exposed to HFD

The growth curve (Figure 1A) shows a positive slope in weight gain for both control and HFD-fed rats. HFD rats have an increased weight from postnatal day (PND) 56 to 62 compared to controls (Interaction: [F_(30, 2046)_ = 14.90; *p* < 0.0001]; Time: [F _(30, 2046)_ = 1300; *p* < 0.0001]; Diet: [F _(1, 2046)_ = 394.7; *p* < 0.0001]). Body weight (Figure 1B; *p* = 0.0065) and retroperitoneal fat (Figure 1C; *p* < 0.0001) at PND 62 were higher in HFD rats compared to those of control rats.

### 2.2. Electrophysiological Recordings in NAcc of Rats Exposed to HFD

Excitatory glutamatergic afferents to NAcc MSNs from the prefrontal cortex are modulated by mesolimbic DA neurons in a frequency-dependent manner [31,32,33]. In addition, rewards such as foods and drugs of abuse increase DA release in NAcc and PFC [9]. However, we do not know if, in our childhood and adolescent obesity model, the glutamatergic synaptic activity is affected in NAcc. First, we classify as putative D_1_-like and D_2_-like MSNs in NAcc which is a form of short-term synaptic plasticity between control and HFD male rats, stimulating the PFC afferents electrical; according to the electrophysiological characterization of MSN described in the dorsal striatum, the D_1_-MSNs and D_2_-MSNs can be identified by exhibiting a non-facilitating and a facilitating response to paired-pulse protocol, respectively (Appendix A) [16,34,35,36,37]. In this work, control and HFD rats showed the two subpopulations of MSNs, which showed a non-facilitation or facilitation by paired-pulse protocol and in turn we named D_1_-like MSNs and D_2_-like MSNs, respectively (Figure 2A; control: (D_1_-like MSNs 0.9270 ± 0.0318, n = 10 and D_2_-like MSNs 1.1450 ± 0.0312, n = 7, *p* = 0.0001) vs. HFD: (D_1_-like MSNs 0.8380 ± 0.0506, n = 9 and D_2_-like MSNs 1.2230 ± 0.0312, n = 9, *p* < 0.0001)). Interestingly, the proportions of D_2_-like MSNs in HFD rats (52.9%) were significantly higher than in control rats (34.5%) (Figure 2A; *p* = 0.0077). Jointly, we found that in basal conditions in overall cells, the frequency of spontaneous excitatory postsynaptic currents (sEPSC) was higher in HFD than in control rats (Figure 2B, basal; HFD 1.395 ± 0.175; n = 13 vs. control 0.718 ± 0.175; n =11; *p* = 0.046, respectively). This finding suggests that the neurotransmitter release or excitability of glutamatergic afferents in HFD rats increases. Then, to assess if the D_1_-like and D_2_-like cells are differentially modified by diet, we separated the sEPSC analysis of D_1_ or D_2_- like MSNs, revealing significant changes in control and HFD frequency, except in control D_2_-like MSNs (Figure 2B, control D_1_-like MSNs, basal 0.652 ± 0.1949, n = 6 vs. amphetamine 0.1898 ± 0.0465, n = 6, *p* = 0.0313; control D_2_-like MSNs, basal 0.7972 ± 0.3313, n = 5 vs. amphetamine 0.2149 ± 0.0622, n = 5, *p* = 0.125; HFD D_1_-like MSNs, basal 0.7904 ± 0.1756, n = 7 vs. amphetamine 0.4048 ± 0.0917, n = 7, *p* = 0.0156; HFD D_2_-like MSNs, basal 2.100 ± 0.6071, n = 6 vs. amphetamine 0.4434 ± 0.1355, n = 6, *p* = 0.313). The increased number of D_2_-like MSNs than D_1_-like MSNs in HFD rats could modify the DA neuromodulation of NAcc synaptic transmission. To test this, we assessed PPR and the EPSC amplitude in response to 0.1 µM amphetamine (AMPH) in control slices and HFD male rats.

In control rats, PPR did not change in the presence of AMPH neither D_1_-like MSNs (before: 0.941 ± 0.037, n = 10 and after: 1.008 ± 0.047, n = 7, *p* = 0.812) nor D_2_-like MSNs (before: 1.134 ± 0.036 and after: 1.198 ± 0.019, n = 5, *p* = 0.125) (Figure 2C, left panel). However, in HFD rats, the PPR of the D_2_-like MSNs population decreased after adding AMPH (before: 1.213 ± 0.026 and after: 1.069 ± 0.064, n = 7, *p* = 0.049), while in D_1_-like MSNs did not change (before: 0.825 ± 0.055 and after: 0.945 ± 0.085, n = 8, *p* = 0.0780) (Figure 2C, right panel), suggesting that AMPH induce a selective increase in glutamate release on the D_2_-like MSNs. Furthermore, the EPSC amplitude of control rats did not change in the presence of AMPH neither D_1_-like MSNs (before: 56.47 ± 13.77 and after: 79.03 ± 18.03, n = 7, *p* = 0.109) nor D_2_-like MSNs (before: 52.31 ± 13.53 and after: 67.53 ± 13.92, n = 5, *p* = 0.625) (Figure 2D, left panel). However, in HFD rats, the EPSC amplitude of the D_2_-like MSNs population increased in the presence of AMPH (before: 41.410 ± 8.083 and after: 78.01 ± 16.76, n = 7, *p* = 0.031), while in D_1_-like MSNs the amplitude remained unchanged (before: 53.86 ± 13.27 and after: 59.41 ± 14.23, n = 8, *p* = 0.740) (Figure 2D right panel). These findings suggest that HFD showed a higher basal excitatory activity of D_2_-like MSNs than D_1_-like MSNs, which is selectively modulated by AMPH-induced DA release, presumably by a heterosynaptic mechanism at pre- and postsynaptic level.

### 2.3. Gene Expression of Neuroinflammatory Markers in NAcc of Rats Exposed to HFD

Fatty acids from diet activate an intracellular protein complex member of nucleotide-binding oligomerization domain-like receptor family (NOD-like) pyrin domain containing 3 (NLRP3) [38]. The activation of NLRP3 recruits the adaptor protein (ASC), leading to the activation of caspase-1, a proteolytic enzyme that promotes the cleavage of proinflammatory cytokines to their mature forms (i.e., IL-1 β, IL-18) (Figure 3A) [39]. The pathophysiological implications of inflammasome activation have been related to developing diseases such as diabetes and obesity, among others [40]. In this context, the chronic exposure to HFD for six weeks increased the expression of *Il-1b* (Figure 3C; *p* = 0.0106), *caspase-1* (Figure 3D; *p* = 0.0428), *Nlrp3* (Figure 3E; *p* = 0.0001), and *Gfap* (Figure 3F; *p* = 0.0003). However, the mRNA levels of ASC were not affected in obese rats (Figure 3B; *p* = 0.8099).

### 2.4. Monoamine Contents in Brain Nuclei of the Mesolimbic and Nigrostriatal Pathways of Rats Exposed to HFD

To evaluate the neurochemical effects of chronic exposure to HFD on DA, serotonin (5-HT), and its primary metabolites, the tissue content of these neurotransmitters was measured using HPLC coupled to electrochemical detection (ED) in NAcc, dorsolateral striatum (DLS), substantia nigra (SN), and VTA. Our data show that both DA (Figure 4A) and 5-HT (Figure 4C) concentration in NAcc, DLS, SN, and VTA were similar between controls and HFD rats. On the other hand, tissue content of 5-hydroxyindoleacetic acid (5-HIAA), a metabolite of 5-HT produced by monoamine oxidase (MAO), was again similar in the micro-dissected brain nuclei of controls and HFD rats (Figure 4D). Interestingly, the content of 3,4-dihydroxyphenylacetic acid (DOPAC), a DA metabolite produced by MAO, was significantly reduced in NAcc (*p* = 0.0247) and DLS (*p* = 0.0052) from obese male rats, without affecting its tissue content in SN and VTA (Figure 3B).

### 2.5. Ex-Vivo DA Release in NAcc of Rats Exposed to HFD

Fast scan cyclic voltammetry (FSCV), an electrochemical technique, measured DA release in NAcc slices with sub-second resolution. Basal and amphetamine-induced DA release stimulated by a single pulse was reduced in NAcc of HFD rats (Figure 5A Basal *p* = 0.0131; 0.1 µM amphetamine (AMPH) *p* = 0.0125). In addition, NAcc DA uptake was decreased in the same conditions (Figure 5B Basal *p* = 0.0317; AMPH *p* = 0.0322). Representative peaks and color plots are observed in panel D, showing a lower peak height (DA release) and DA oxidation in NAcc of rats exposed to HFD for six weeks (Figure 5D).

NAcc DA release evoked by phasic five-pulse stimulation was higher at 20, 40, and 100 Hz frequencies in HFD rats compared to control rats (Figure 5C) (Interaction: [F(4, 65) = 0.4804; *p* = 0.7500]; frequencies: [F (4, 65) = 25.87; *p* < 0.0001]; Diet: [F (1, 65) = 18.85; *p* < 0.0001]).

## 3. Discussion

This work aimed to evaluate the synaptic, neurochemical, and molecular changes occurring in NAcc of male rats fed with HFD for six weeks (from PND 21 until PND 62). This period is a stage of infant-juvenile growth where the animals have sustained growth associated with a great anabolism of the musculoskeletal system. However, despite this period of high energy expenditure in our animal model, we can observe a significant increase in body weight and adipose tissue in rats exposed to HFD (Figure 1).

In humans, the development of the reward system is established from the age of one to eight, while in rodents, it develops from the first to the fifth postnatal week [25]. Therefore, exposure to HFD during this period could strongly affect the reward system and the hedonic eating control, which is still in development. In this context, the administration of highly palatable foods such as chocolate, sweetened beverages, and snacks increases DA release in NAcc in adult rodents [41,42,43], leading to an increase in dopaminergic neurotransmission, similar to other stimuli such as drugs of abuse [13]. Indeed, chronic exposure to obesogenic foods produces behavioral and neurobiological aspects identical to drug addiction [44,45].

Interestingly, using positron emission tomography in humans, an increase in the availability of D_2_ was demonstrated in the ventral striatum, putamen, and midbrain of the obese compared to normal-weight subjects [46]. Exposure to a high-fat and sugar diet for 12 weeks in adult mice reduces D_1_ and increases D_2_ expression in NAcc [47]. In this sense, our electrophysiological data suggest that in HFD, the putative D_2_-like MSNs showed a higher basal glutamatergic activity which AMPH downregulates, than that of chow which was our control group. This decrease in sEPSC frequency induced by AMPH occurs along with the increase of glutamate release probability (i.e., PPR; Figure 2C,D) [31,48]. These contradictory AMPH effects could be explained by indirect activation of D_2_ receptors that (1) downregulate the firing rate of action potentials (AP) on the glutamatergic terminals (i.e., decreasing the sEPSC frequency) and (2) inhibiting the lateral inhibition on glutamatergic and GABAergic terminals on MSNs. However, the regulation by hetero-receptors on cortical and dopaminergic inputs in NAcc (e.g., endocannabinoid, acetylcholine, and adenosine, among others) is an open question in our work [33,49].

Our findings suggest that in obese rats, the high level of glutamatergic activation of the indirect pathway (D_2_-like MSNs) compared to the direct pathway (D_1_-like MSNs) could be related at the slightest basal DA release observed in this work (Figure 5). Our results suggest that this imbalance could trigger an abnormal form of hetero-synaptic plasticity, which is sufficient to modify the eating behavior. However, to identify the receptors implicated in these HFD-induced electrophysiological changes, and these should be determined with further pharmacological and molecular experiments. Finally, the classical implications associated with the change in the activity of the D_1_-MSNs and D_2_-MSNs, or the changes in the expression patterns of D_1_ and D_2_ receptors have been related to exclusive effects on direct or indirect pathways in NAcc. However, this topic is being reconsidered, since studies that have combined electrophysiological, optogenetic, and chemogenetic techniques have shown in NAcc that D_1_-MSNs innervate the ventral midbrain (direct pathway) and the ventral pallidum (indirect pathway), which receives closely about 90% and 50% of D_2_-MSNs and D_1_-MSNs afferents, respectively [19,50].

During the last decade, several studies have suggested that obesity is associated with an inflammatory process characterized by increased plasma levels of proinflammatory cytokines such as TNF-α, IL-6, and C-reactive protein [51]. The inflammation of the nervous system or neuroinflammation induced by foods has been observed in developed societies that have adopted new styles of eating based on ultra-processed foods rich in carbohydrates and fats and low content in vegetables, fiber, and prebiotics [52,53]. Obesogenic foods promote hyperphagia and other behaviors related to malnutrition, facilitating the development of obesity and neuroinflammation [54]. For example, foods with a high glycemic load cause an increase in reactive oxygen species that favor the expression of proinflammatory genes [55]. In this context, inflammation is associated with the activation of an intracellular protein complex called inflammasome, which is assembled in response to damage-associated molecular patterns (DAMP) and pathogen-associated molecular patterns (PAMP) [40]. NLRP3 inflammasome can be activated by cholesterol crystals and ceramides [56], products generated due to the exacerbation of lipolysis in obesity [57]. The NLRP3 inflammasome is made up of a sensor protein (NLRP3), an adapter molecule ASC (apoptosis-associated speck-like protein), and the effector enzyme pro-caspase 1 [58]. The inflammasome activation favors the proteolytic cleavage of Pro-IL-1β and Pro-IL-18 by caspase 1, generating the respective active proteins (IL-1β and IL-18), which promote the inflammatory response [59,60]. Furthermore, IL-1β is a critical pathological mediator of obesity-induced insulin resistance [61], and NLRP3 expression depends on NF-κB [61]. Our results show that chronic exposure to HFD is enough to increase the expression of inflammasome components and GFAP (a marker of astrogliosis) in NAcc (Figure 3). In this context, it has been shown that exposure to HFD increases the NAcc expression of proinflammatory cytokines, NF-κB, Iba-1, and GFAP, respectively [30]. In addition, the increase of proinflammatory cytokines affects the dopaminergic neurochemistry in NAcc, since systemic administration of IL-6 and IL-2 decrease NAcc DA extracellular levels [62]. In our model of chronic HFD exposure, the increase in neuroinflammatory markers in NAcc (Figure 3) may be part of the pathophysiological mechanism associated with the decreased basal levels of NAcc DA observed using FSCV ex vivo in obese animals (Figure 5A,C). It has recently been shown using FSCV ex vivo that exposure to HFD from postnatal days 42 to 84 decreases phasic DA release (5 pulses 20 Hz) and reuptake rate in NAcc slices from mice of both sexes [63]. Interestingly, after restoring baseline values of tonic (monophasic) DA release, the perfusion of ACSF plus IL-6 (10 nM) or ACSF plus TNFα (300 nM) significantly reduces NAcc DA release in mice fed with HFD [63].

Regarding the neurochemical changes, we observed that the total DOPAC content in NAcc and DLS decreased in male rats exposed to HFD for six weeks (Figure 4B). DOPAC is formed through oxidation mediated by the enzyme MAO expressed at the presynaptic terminal and the extracellular level in glial cells. A reduction in DOPAC content may reflect a decrease in DA content (not observed in our work) or a decrease in DA metabolism. On the other hand, a reduction in DAT expression mediated by chronic exposure to HFD may be responsible for the decline in DA reuptake and its presynaptic metabolism. This hypothesis may be supported by results showing that exposure to HFD for 20 days in 12-month-old adult mice increases D_2_ binding in the striatum and NAcc shell while DAT binding decreases in the identical nuclei [64]. In addition, in synapto-neurosomes obtained from adult male rats exposed for one month to ad libitum food, food restriction, or obesogenic diets, it was observed that striatum DA reuptake was lower in obese rats. It was associated with a decrease in the binding of [^3^H]-CFT (a radiotracer of DAT) and a decrease in DAT expression in striatum membranes [65]. Our data of ex vivo FSCV show a reduction in basal DA release in NAcc slices of rats exposed to HFD (Figure 5A and C). This reduction in NAcc DA release could be due to an increase in D_2_ expression and turn, reduced DA release through an auto-receptor mechanism (Figure 5A). In this sense, exposure to HFD plus sugar for 3 to 4 weeks in rats decreased DA release in NAcc core and striatum [66].

Our data show that perfusion of NAcc slices of obese rats with (0.1 µM) AMPH did not produce a statistically significant change in DA extracellular levels (Figure 5A), possibly it is due to HFD-induced decreased DAT expression (pharmacological target of AMPH). It has been shown that exposure to HFD for six weeks in 2-month-old rats reduces DA reuptake in vivo and DAT expression in ventral striatal membranes [67]. Mice that had access for three hours daily, during three days per week, for six weeks in total, presented a high preference for the HFD, an increase of phasic DA release, and a reduction in amphetamine-induced DA uptake inhibition compared to control animals exposed to food chow for six weeks [68]. On the other hand, 42-day-old mice of both sexes fed with HFD for six weeks did not show changes in single-pulse-evoked tonic DA release and the reuptake rate [69].

## 4. Materials and Methods

### 4.1. Reagents

DA, 5-HT, DOPAC and 5-HIAA standards, EDTA, and 1-octanesulfonic acid were purchased from Sigma-Aldrich, Inc. (St. Louis, MO, USA). AMPH sulfate was obtained from Laboratorio Chile S.A. (ISPCH No F-1386/18, Ñuñoa, Santiago, Chile). All other reagents were of analytical and molecular grade. Chow diet (Prolab^®^ Isopro^®^ RMH 3000, St. Louis, MO, USA) and HFD (D12492 Research Diets^®^, New Brunswick, NJ, USA) were purchased from Animal Care (San Joaquín, Santiago, Chile) and PrionLab (Peñaflor, Santiago, Chile), respectively.

### 4.2. Animals

Sixty-eight male Sprague-Dawley rats from different litters were considered for the following experimental groups: control (n = 32) and HFD (n = 36). Animals from the vivarium of the Faculty of Science of the Universidad de Valparaíso were used and housed in a temperature- and humidity-controlled room (22 ± 2 °C; 50 ± 5%, respectively) under artificial illumination (12-h light/12-h dark; light on at 08:00 a.m.), with food and water ad libitum from postnatal (PND) 21 to 62 (Figure 6). Control and HFD male rats were fed with a standard chow diet (calories provided by 26% protein, 14% fat, and 60% carbohydrates) and HFD (calories supplied by 20% protein, 60% fat, and 20% carbohydrates), respectively. Efforts were made to minimize the number of rats used and their suffering.

### 4.3. Experimental Procedure

After weaning PND21, the experimental group-housed rats (n = 3–4) in standard cages under vivarium conditions until PND62. At PND62, rats were anesthetized with isoflurane (5% in 0.6 L/min air flow) in an induction chamber using an animal anesthesia system (model 510, RWD Life Science Co. Ltd., Shenzhen, China). When the rats were deeply anesthetized, they were euthanized by decapitation with a guillotine (model 51330, Stoelting^TM^ Co., Wood Dale, IL, USA). Their brains were quickly removed for electrophysiological, molecular, and neurochemical experiments. Peripheral tissues such as retroperitoneal fat (Figure 1C), tibial muscle, and serum were collected from each animal.

### 4.4. Electrophysiological Studies: Slice Preparation, Recording, and Analysis

A total of 11 control and 13 HFD male rats were used. After deep anesthesia with isoflurane, the animals were euthanized by decapitation with a guillotine. Their brains were quickly removed and placed into ice-cold (4 °C) artificial cerebrospinal fluid (ACSF), concentrations (mM) of which were: 124.0 NaCl, 2.7 KCl, 1.25 KH_2_PO_4_, 2.0 Mg_2_SO_4_, 26.0 NaHCO_3_, 2.5 CaCl_2_, and 10.0 glucose. ACSF was bubbled with carbogen gas (95% O_2_; 5% CO_2_; pH 7.4; Linde Gas Chile S.A.), and brains were cut in coronal slices (300–350 µm) using a vibrating Microtome (model vibroslice VSL; World Precision Instruments, Sarasota, FL, USA). Slices were incubated in ACSF for one hour at room temperature before electrophysiological recordings began. Slices were transferred to an immersion-recording chamber (2 mL), fixed to an upright microscope stage (model FN100 IR; Nikon Inc., Tokyo, Japan) equipped with infrared and differential interference contrast imaging devices and with a 403-water immersion objective. Whole-cell currents and voltage-clamp recordings were performed from MSNs within the NAcc core, identified under visual guidance using infrared video microscopy, and based on cell soma size and firing properties [2]. Whole-cell voltage and current clamp recordings using an amplifier (model EPC-7, Heka Instruments, Germany) were made from MSN voltage clamped at −60 mV, using patch-type pipette electrodes (3–5 MΩ) containing (in mM): 135.0 KMeSO_4_, 10.0 KCl, 10.0 N-(2-hydroxyethyl)-1-piperazine-ethanesulphonic acid (HEPES), 5.0 NaCl, 5.0 ATP-Mg^2+^ and 0.4 GTP-Na^+^ (pH = 7.2). eEPSCs and sEPSC were recorded and imported as HEKA pulse data by Microcal Origin 6.0 software to be analyzed offline and graphed, using analysis software (Clamfit 6.0, Molecular Devices LLC., San Jose, CA, USA).

Experiments started after a 5–10 min stabilization period following the establishment of whole-cell configuration. In voltage clamp mode, evoked excitatory postsynaptic currents (eEPSCs) were recorded, filtered at 3.0 kHz, acquired at 4.0 kHz using an A/D converter (model ITC-16, InstruTECH, Reutlingen, Germany), and stored with Pulse FIT software (Heka instruments, Germany). Cells that exhibited a significant change in access resistance (>20%) were excluded from the analysis. Stimulation of mesolimbic pathways (200 ms duration, 2.0 s^−1^) was performed by introducing two silver chloride electrodes into an ACSF-filled, patch-like pipette made with septum theta capillaries (World Precision Instruments, Sarasota, FL, USA). The stimulation pipette was gently placed medial line around the NAcc core at 200–300 µm to the dendritic tree of the recorded cell (<200 mm) and fixed when a single response was detected. Single stimuli (60 s, 50–100 ms, 20–100 mA; Master 8, AMPI, Israel) through an isolation unit (Isoflex, AMPI, Israel). eEPSCs were recorded and analyzed offline using analysis software (Clamfit 6.0, Molecular Devices LLC., San Jose, CA, USA). Paired-pulse protocol and calculated the paired-pulse ratio (PPR) to estimate putative presynaptic changes. PPR was calculated as (R2/R1), where R1 and R2 are the peak amplitudes of the first and second eEPSC (80 ms apart). The mean eEPSC amplitude values and PPR were obtained for each condition.

### 4.5. RT-qPCR

A total of 10 control and 11 HFD male rats were used. Rats were decapitated, and their brains were removed quickly. NAcc and other brain areas were micro-dissected at 4 °C using a micro-punch, weighed on an analytical balance, and stored at −80 °C for further analysis. RT-qPCR was used to determine relative expressions of *Asc*, *Il-1β*, *caspase-1*, *Nlrlp3*, and *Gfap* in NAcc of control and HFD rats. Total RNA was extracted using the E.Z.N.A.^®^ Total RNA Kit I (Cat. No R6834-02; Omega BioTek Inc., Norcross, GA, USA) according to the manufacturer’s instructions. RNA was quantified using the microplate Spectrophotometer (model Epoch; BioTek Inc., Winooski, VT, USA), and RNA integrity was assessed through agarose gel electrophoresis. Total RNA from each sample was reverse transcribed with the RevertAid RT kit (Cat. N° K1691, Thermo Fisher Scientific, Waltham, MA, USA), according to the manufacturer’s instructions. Real-time RT-PCR was performed using Supermix SsoAdvanced Universal SYBR Green (Cat. N° 1725271, Bio-Rad, Hercules, CA, USA) by the manufacturer’s instructions. For specific gene amplification, a standard protocol of 40 cycles was used in a Real-Time PCR Detection System (model CFX96™; Bio-Rad, Hercules, CA, USA). Ribosomal 18S mRNA was measured in each protocol to normalize the gene expression using primers reported previously [70,71]. The details of the primers used for each analyzed gene are summarized in the following Table 1.

The specificity of each generated amplicon was confirmed by performing melting curves at the end of each reaction. Results were expressed as fold change by the 2^ΔΔCT^ method [72].

### 4.6. Neurochemical Studies

A total of 11 control and 12 HFD male rats were used to analyze monoamine content and ex vivo fast-scan cyclic voltammetry experiments.

#### 4.6.1. Monoamine Content Quantification Using HPLC-ED

A total of 4 control and 4 HFD male rats were used. Rats were anesthetized with isoflurane (3% in 0.8 L/min air flow) in an induction chamber for 3 min and decapitated. NAcc, DLS, SN, and VTA were at 4 °C using a brain matrix (model 68711; RWD Life Science, Shenzhen, P.R. China) and micro-punch (model 15076 Harris Uni-Core; Ted-Pella Inc., Redding, CA, USA). Brain nuclei were weighed on an analytical balance (model JK-180; Chyo balance corp, Tokyo, Japan) and homogenized in 400 μL of 0.2 M perchloric acid using a sonicator (model Q55; Qsonica, Newtown, CT, USA). The homogenate was centrifuged to 12,000× *g* for 10 min at 4 °C (model Z233MK-2; Hermle Labor Technik GmbH, Wehingen, Germany), and the supernatant was filtered using HPLC syringe filters (model EW-32816-26; Cole-Parmer, Vernon Hills, IL, USA). Ten microliters of final supernatant were injected into HPLC-ED with the following configuration: Isocratic pump (model PU-2080 Plus; Jasco Co. Ltd., Tokyo, Japan), C18 column (model Kromasil 100-3.5-C18; AkzoNobel, Bohus, Sweden), and electrochemical detector (model LC-4C; Bioanalytical System Inc., West Lafayette, IN, USA) set at 0.650 V (oxidation potential), 0.5 nA (sensitivity), and 0.03 Hz (electrical noise). The composition of the mobile phase was 0.1 M NaH_2_PO_4_, 1.5 mM 1-octanesulfonic acid, 1.28 mM EDTA, 2.0% (^v^/_v_) tetrahydrofuran, and 4.5% (^v^/_v_) CH_3_CN (pH 4.0). It was pumped at a flow rate of 0.105 mL/min. DA (18.6 min), DOPAC (14.2 min), 5-HIAA (22.3 min), and 5-HT (48.4 min) content were assessed by comparing the respective peak area and elution time of the sample with a reference standard. The quantification was performed using a calibration curve for each neurotransmitter and metabolite (Program ChromPass, Jasco Co. Ltd., Tokyo, Japan). The concentration was expressed as pg per mg of wet tissue.

#### 4.6.2. Ex-Vivo Fast Scan Cyclic Voltammetry

A total of 7 control and 8 HFD male rats were used. Rats were anesthetized with isoflurane (3% in 0.8 L/min air flow) in an induction chamber for 3 min and decapitated. The brain was quickly removed and sliced (300–350 µm) using a vibrating Microtome (model vibroslice VSL; World Precision Instruments, Sarasota, FL, USA). Brain slices were obtained in ice-cold (4 °C) artificial cerebrospinal fluid (ACSF), whose composition was: 126.0 mM NaCl, 25.0 mM NaHCO_3_, 11.0 mM glucose, 2.5 mM KCl, 2.4 mM CaCl_2_, 1.2 mM MgCl_2_, 1.2 mM NaH_2_PO_4_ and 0.4 mM L-ascorbic acid (adjusted to pH 7.4). ACSF was bubbled with carbogen gas (95% O_2_; 5% CO_2_; pH 7.4; Linde Gas Chile S.A.). A glassy-carbon microelectrode (working electrode) was linearly scanned (−0.4 to 1.2 V and back to −0.4 V vs. Ag/AgCl). Cyclic voltammograms were assessed at the carbon fiber electrode every 100 ms with a scan rate of 400 V/s using a voltammeter/amperemeter (model Chem-Clamp Potentiostat, Dagan Corporation, Minneapolis, MN, USA). In this condition, DA is oxidized at +0.6 V and reduced at −0.4 V. To measure NAcc DA release, a concentric bipolar electrode (model 30200; FHC, Bowdoin, ME, USA) was stimulated with the following parameters: monophasic, 4 ms and 400 μA (current stimulus isolator NL800A; Digitimer, Ltd., Hertfordshire, UK) every 3 min. In addition, an Ag/AgCl reference electrode (model EP2; World Precision Instruments, Sarasota, FL, USA) was used. Phasic DA release was measured using 400 μA of stimulation and trains of 5 pulses at 5, 10, 20, 40, and 100 Hz. For data collection, two National Instruments acquisition cards (NI-DAQ; PCI-6711 and PCI-6052e; National Instruments, Austin, TX, USA) were used to interface the potentiostat and stimulator with Demon Voltammetry and Analysis software (Wake Forest Health Sciences, Winston-Salem, NC, USA) [73]. After phasic stimulation, DA release was measured again with monophasic stimulation described above. AMPH (0.1 μM) dissolved in ACSF was applied to NAcc slices. After each experiment, working electrodes were calibrated using ACSF containing 3 μM DA.

### 4.7. Statistical Analysis

Data were expressed as mean ± SEM. Two-way ANOVA followed by multiple comparisons (Bonferroni post-hoc test) was used to evaluate changes in body weight (Figure 1A) and multi-pulse DA release (Figure 5B). To compare analysis in the spontaneous EPSC frequency, we used *t*-test Welch two-tailed, and for differences before and after drug for PPR and evoked EPSC amplitude, we used *t*-test Mann–Whitney and Wilcoxon two-tailed (Figure 2). Figure 1A–C, Figure 3B–E,G–K, Figure 4A–D and Figure 5A were analyzed using unpaired *t*-test. Statistical analyses were conducted with GraphPad Prism v9.4.1 (GraphPad Software, San Diego, CA, United States), and *p* < 0.05 was considered statistically significant.

## Figures and Tables

**Figure 1 ijms-24-04703-f001:**
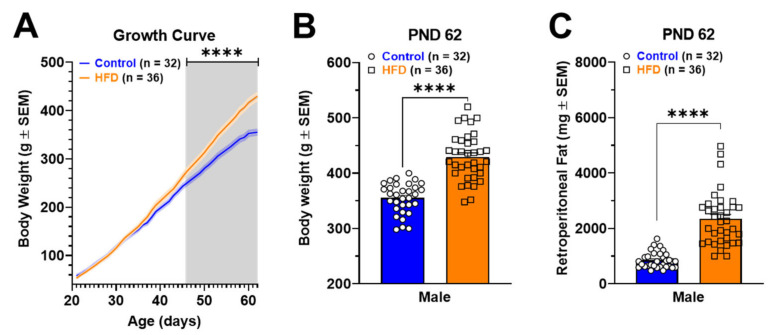
Effects of exposure to HFD from postnatal day (PND) 21 to 62 in male rats. Panel A shows the growth curve during six weeks of male control and HFD rats. Panels B and C show the body weight and retroperitoneal fat pad at the end of the experiments (PND 62). The statistical analysis was a two-way ANOVA multiple comparisons test (**A**) and unpaired *t*-test (**B**,**C**) (**** *p* < 0.0001).

**Figure 2 ijms-24-04703-f002:**
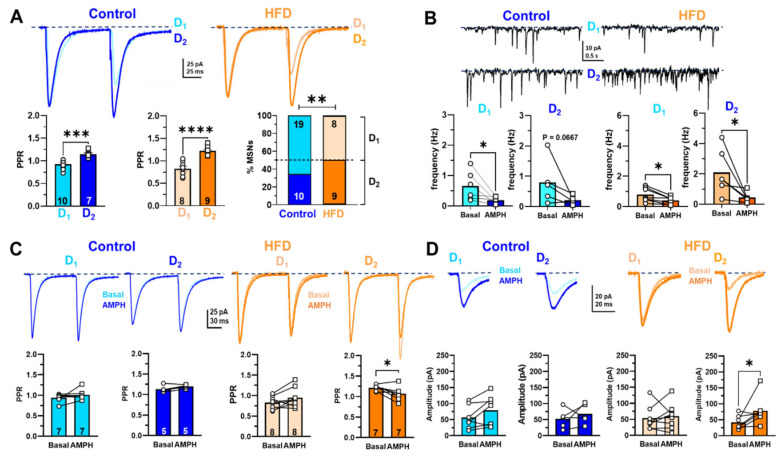
Changes in the NAcc glutamatergic synaptic transmission of control and HFD male rats. (**A**) Representative PPR average traces were recorded in D_1_-like MSNs and D_2_-like MSNs of the control group (light and dark blue, respectively) and in D_1_-like MSNs and D_2_-like MSNs of HFD (light and dark orange, respectively). Summary graphs of PPR average values in control (**left)** and HFD (**middle**) groups, showing depression in D_1_-like MSNs and facilitation in D_2_-like MSNs in both groups. The proportion of D_1_-like and D_2_-like cells in control and HFD groups (**right**). (**B**) Representative traces of spontaneous synaptic activity were recorded in D_1_ (**upper** traces) and D_2_ (**lower** traces)-like MSNs of control (**left**) and HFD (**right**) in basal conditions. The lower panel shows the summary graphs of spontaneous EPSC frequency in basal and AMPH conditions (0.1 μM) for two cell types, comparing the mean values between groups. (**C**) Superimposed PPR average traces were recorded in representative D_1_-like MSNs (**upper**) and D_2_-like MSNs (**lower**) control cells in basal and in the presence of AMPH (light and dark blue, respectively). Summary graphs of PPR in control (**left**) in D_1_-like and D_2_-like cells, comparing the average value in basal and AMPH conditions. Summary graphs of PPR in HFD (**left**) in D_1_-like and D_2_-like cells, comparing the average value in basal and AMPH conditions. (**D**) Average EPSC evoked by a single pulse recorded in single D_1_-like and D_2_-like cells of control rats, superimposing the traces recorded in basal and AMPH conditions (**upper** panel). Summary graphs of average values of EPSC amplitude in control in D_1_-like and D_2_-like cells, comparing the average value in basal and AMPH conditions (**lower** panel). Average EPSC evoked by a single pulse recorded in single D_1_-like and D_2_-like cells of HFD rats, superimposing the traces recorded in basal and AMPH conditions (**upper** panel). Summary graphs of average values of EPSC amplitude in D_1_-like and D_2_-like cells, comparing the average value in basal and AMPH conditions (**lower** panel). Data are expressed as means ± SEM (mean standard error), and the statistical analysis used were *t*-test Welch one-tailed (Figure 2A), Mann–Whitney two-tailed two-way (Figure 2B), and Wilcoxon two-tailed (Figure 2C,D) (* *p* <0.05; ** *p* <0.01; *** *p* <0.001; **** *p* <0.0001).

**Figure 3 ijms-24-04703-f003:**
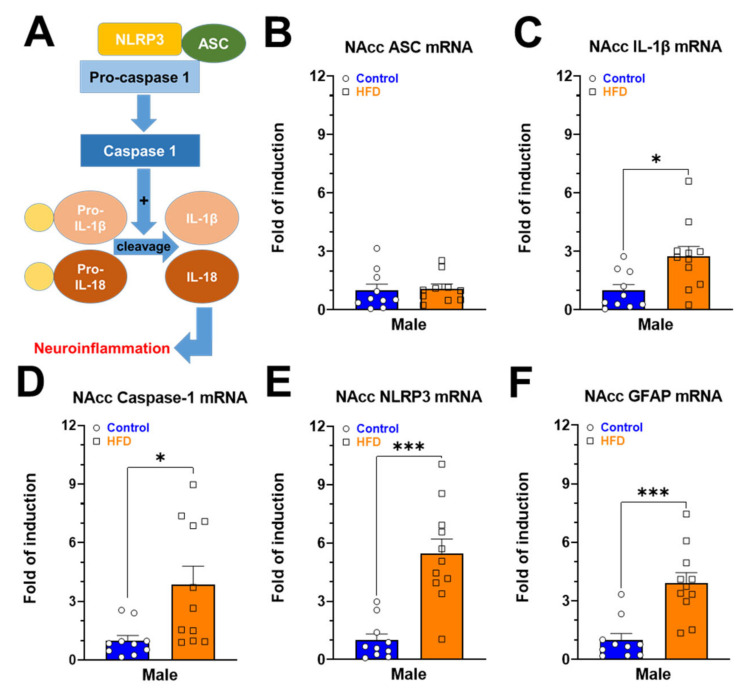
Gene expression patterns of neuroinflammatory markers (**A**–**F**) in NAcc of normal weight and obese rats. All data are normalized for 18S expression levels within the same sample. Results are expressed as fold induction relative to the control group, and the results were analyzed by an unpaired *t*-test (* *p* <0.05; *** *p* <0.001).

**Figure 4 ijms-24-04703-f004:**
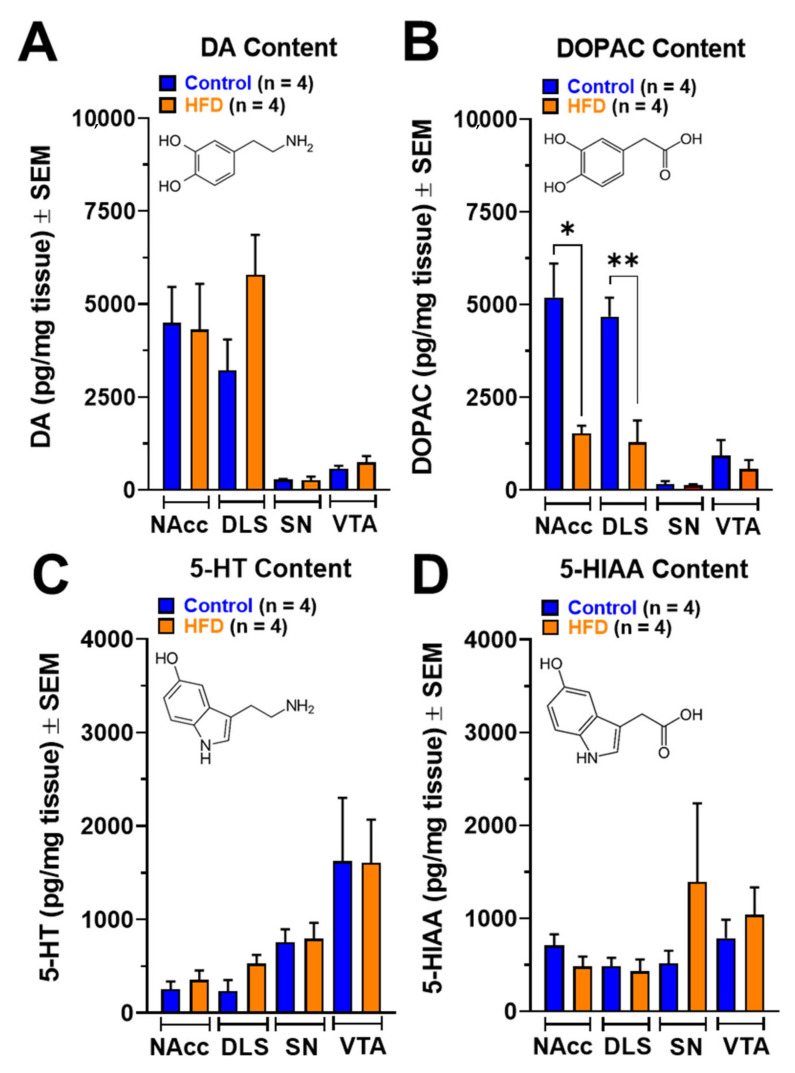
DA (**A**), DOPAC (**B**), 5-HT (**C**), and 5-HIAA (**D**) contents in brain nuclei of the mesolimbic (NAcc and VTA) and nigrostriatal (DLS and SN) pathways of rats exposed to HFD for six weeks. Results are expressed as pg per mg of tissue and were analyzed by unpaired *t*-test (* *p* < 0.05; ** *p* < 0.01).

**Figure 5 ijms-24-04703-f005:**
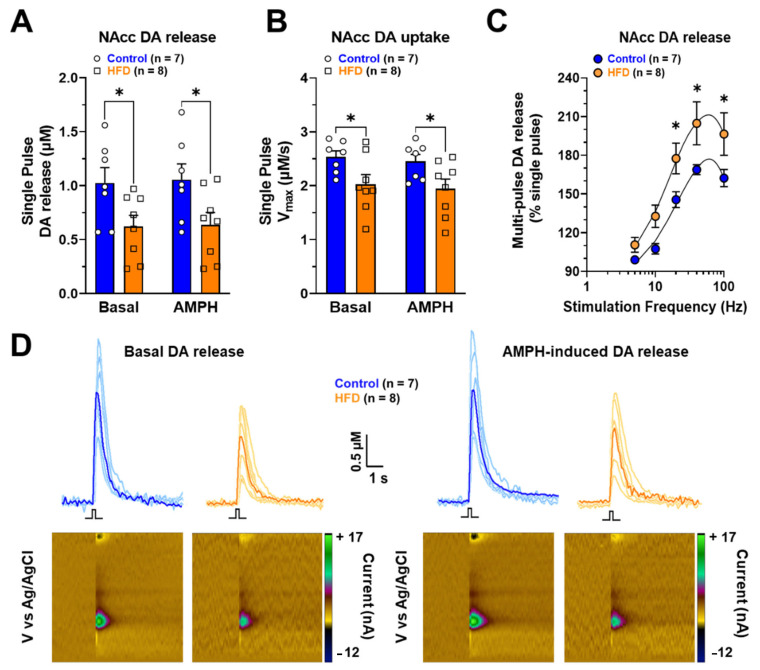
NAcc DA release in slices of normal weight and obese male rats. (**A**) NAcc DA release stimulated by a single pulse in basal condition and with 0.1 µM amphetamine (AMPH). (**B**) Maximal rate of DA uptake (Vmax) in NAcc. (**C**) Phasic DA release in NAcc stimulated by five pulses at 5, 10, 20, 40, and 100 Hz frequencies. (**D**) Representative line traces showing peak height (DA release) and representative color plots in a slice of rats fed with chow or HFD for six weeks. Blue and orange represent the mean DA release in control and HFD male rats, respectively. The statistical analysis used was a two-way ANOVA multiple comparisons test (Figure 5A) (* *p* < 0.05). Monophasic stimulation was 400 μA during 4 ms.

**Figure 6 ijms-24-04703-f006:**
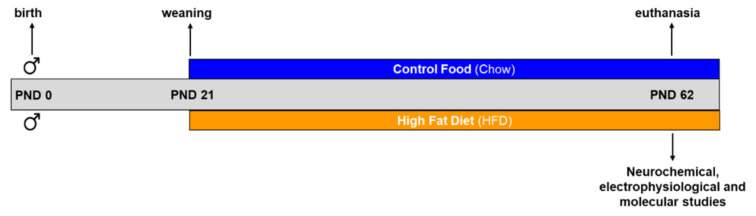
Schematic description of the exposure to diets, the most important milestones, and the experiments carried out.

**Table 1 ijms-24-04703-t001:** Description of genes analyzed, indicating the GenBank accession number and the primer sequences.

Gene	GenBank Access (N°)	Forward Primer	Reverse Primer
**ASC**	NM_172322.1	5′-TGCTGCAGATGGACCCCATA-3′	5′-CACAGCTCCAGACTCTTCCATA-3′
**IL-1β**	NM_031512.2	5′-AGCTTCAGGAAGGCAGTGTC-3′	5′-TCAGACAGCACGAGGCATTT-3′
**Caspase-1**	NM_012762.3	5′-CACGAGACCTGTGCGATCAT-3′	5′-CTTGAGGGAACCACTCGGTC-3′
**NLRP3**	NM_001191642.1	5′-CTCTGCATGCCGTATCTGGT-3′	5′-GTCCTGAGCCATGGAAGCAA-3′
**GFAP**	NM_017009.2	5′-CAACCTCCAGATCCGAGAAACC-3′	5′-GCATCTCCACCGTCTTTACCA-3′
**18s**	NR_046237.2	5′-TCAAGAACGAAAGTCGGAGG-3′	5′-GGACATCTAAGGGCATCACA-3′

## Data Availability

All data are included in this work.

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
