# Peer review of "Chronic Exposure to High Fat Diet Affects the Synaptic Transmission That Regulates the Dopamine Release in the Nucleus Accumbens of Adolescent Male Rats"

_ijms, 2023, doi:10.3390/ijms24054703_

Round 1
Reviewer 1 Report
Major points
1. Line 135-136= “Interestingly, the percentage of D2-like MSNs tends to increase in HFD rats (from 41% 135 to 53% (Fig. 1B)”. Are the authors referring to Fig 2B?. Moreover, I have concerned regarding this state without a proper immunohistochemical characterization of the number of D1 vs D2 in HFD and Lean. Moreover, I suggest to characterize via immunohistochemistry the number of excitatory puncta on D1 vs D2 to support the electrophysiological data.
2. I suggest to split the image in Fig 2 and showing the analysis of sEPSC dividing D1 and D2 to be consistent with the PPR experiments.
3. Usually, the analysis of Paired pulse ratio if made in a inter-stimulus interval (ISI) ranging from 10ms up to more than 1 second. In this work authors show the results only at 80ms. The author should provide the dynamic of release at a greater range of ISI or justify the use of only 1 ISI.
Minor Poins
1. Title: Although neuroimaging studies showed that certain high palatable foods have effects on the human brain compared to those of addictive drugs leading to symptoms of addiction-like behaviour, food addiction is still a debated topic, see Greenberg, D. & St. Peter, J. V. Sugars and Sweet Taste: Addictive or Rewarding? IJERPH 18, 9791 (2021). I suggest to review the title.
2. Fig 2 C HFD: There is some artifact in the traces respect the other traces showed
3. I suggest to reduce the interval in the Y axis of the figures: ex 2A, 2D, 3B-C to improve the visualization of the data
4. I suggest to move all the mean and SEM values in the figures caption in order to improve the text usability.
5. Line 257: I suggest to modify “show” with “receives a higher basal glutamatergic activity…”
Author Response
REVIEWER Nº1: REPORT FORM
Major points
- A) Line 135-136= “Interestingly, the percentage of D2-like MSNs tends to increase in HFD rats (from 41% 135 to 53% (Fig. 1B)”. Are the authors referring to Fig 2B?.
ANSWER 1A
Thank you for this comment. We made a mistake, and effectively we refer to Fig 2B., which was corrected in the text. The error was fixed in the text (highlighted in green; Page 3, lines 124 - 126).
- B) Moreover, I have concerned regarding this state without a proper immunohistochemical characterization of the number of D1 vs D2 in HFD and Lean. Moreover, I suggest to characterize via immunohistochemistry the number of excitatory puncta on D1 vs D2 to support the electrophysiological data.
ANSWER 1B
We appreciate the reviewer's comment. In this context, one of our aims was to carry out an electrophysiological description in the Nucleus Accumbens of rats exposed to a model of childhood obesity, regard to control animals. For this reason, we wanted to be cautious in mentioning "D1-like MSNs and D2-like MSNs" in the text, according to electrophysiological criteria published in the literature on this topic. However, in future work, we want to fine-tune the immunohistochemistry or immunofluorescence technique to identify the type of MSN neurons effectively. We regret that we cannot carry out this type of experiment.
2) I suggest to split the image in Fig 2 and showing the analysis of sEPSC dividing D1 and D2 to be consistent with the PPR experiments.
ANSWER 2
We fully agree with the reviewer's comment. Since most of the sEPSC frequency results in the current Figure 2A were recorded in the same cells as shown in Figure 2B, we have modified figure 2, the figure 2 caption (highlighted in yellow; Page 5, lines 161 -170), and the description of the results (highlighted in yellow; Pages 3 - 4, lines 120 -140).
3) Usually, the analysis of Paired pulse ratio if made in a inter-stimulus interval (ISI) ranging from 10ms up to more than 1 second. In this work authors show the results only at 80ms. The author should provide the dynamic of release at a greater range of ISI or justify the use of only 1 ISI.
ANSWER 3
We appreciate the reviewer's comment in this context, based on D1 and D2 characterization made by Cepeda et al. (2008), where PPRs were measured at 20 and 10 Hz (i.e., ISI at 50 and 100 ms), we decided to use a range of 80 ms to evaluate depression or facilitation. This range has been used and previously published to determine these properties in various brain structures (Bolshakov and Siegelbaun, 1995; Hsia et al., 1198; Kreitzer and Malenka, 2005; Bonansco et al., 2011; Wang et al., 2012).
We include a brief description about it (highlighted in green; Page 13, lines 412 -414).
Minor Points
4) Title: Although neuroimaging studies showed that certain high palatable foods have effects on the human brain compared to those of addictive drugs leading to symptoms of addiction-like behaviour, food addiction is still a debated topic, see Greenberg, D. & St. Peter, J. V. Sugars and Sweet Taste: Addictive or Rewarding? IJERPH 18, 9791 (2021). I suggest to review the title.
ANSWER 4
We appreciate the reviewer's comment, and considering this suggestion, we have decided to delete the final part about "...: Implications in Food Addiction" from the title.
5) Fig 2 C HFD: There is some artifact in the traces respect the other traces showed.
ANSWER 5
We fully agree with the reviewer's comment. We change these record traces by others with less noise (Please, see new figure 2).
6) I suggest to reduce the interval in the Y axis of the figures: ex 2A, 2D, 3B-C to improve the visualization of the data
ANSWER 6
We appreciate the reviewer's comment. We change the range of the Y axis for a more appropriate one that allows visualizing the results better (Please, see new figures 2 and 3).
7) I suggest to move all the mean and SEM values in the figures caption in order to improve the text usability.
ANSWER 7
Thank you for this comment. We have removed the description of mean ± SEM from all figure captions.
8) Line 257: I suggest to modify “show” with “receives a higher basal glutamatergic activity…”
ANSWER 8
We appreciate the reviewer's comment. We have made the verb change to clarify the sentence better (highlighted in yellow; Page 9, lines 260 -261).

Reviewer 2 Report
This manuscript by Plaza-Briceno et al. evaluates the effect of a high fat diet on several diverse measures of glutamatergic synaptic signaling, inflammation, and dopaminergic signaling in the nucleus accumbens using a wide range of applicable techniques. The study has a variety of potentially interesting results and does stand out for its diversity of approaches. However, in current form it also suffers from some significant weaknesses as further outlined below.
Major comments:
One the cell physiology (Fig. 2):
-Relative to options broadly available, PPR is not a particularly robust way to identify D1 vs. D2 MSNs. At a minimum, it would be reassuring to see PPR correctly predicts sensitivity to a selective D2 agonist in the current animal model.
-It seems unlikely that the ratio of D1 to D2 MSNs identified by PPR in control vs. HFD animals (10:7 vs. 8:9) is significantly different. This question should be evaluated with appropriate statistical tests and de-emphasized, reinterpreted, or removed if there is no significant difference.
-It’s not entirely clear if the spontaneous EPSC data in Fig. 2A was collected in an entirely different group of cells than the evoked EPSC data in the rest of the figure. It seems it would be possible to collect all of these data in the same set of cells if the sweep length was long enough to collect spontaneous data after the evoked responses. If this is in fact what was done, then it would be interesting to know if spontaneous EPSC frequency differed in presumed D1 vs. D2 MSNs.
-Similarly, are the single pulse experiments in Fig. 2D an analysis of the first of the paired pulses in Fig. 2C, or are results in these panels from separate experiments in a separate group of cells? Either is acceptable, but text should be clear on this point.
-While I believe separating results by D1 vs. D2 is still highly valuable, recent work has begun to shed some doubt on the canonical idea that these receptors are good markers of the direct vs. indirect pathways (e.g. see PMIDs 26214370 and 30622165), and I think the authors should probably consider this in the discussion.
-In the discussion (line 257) the authors state that results indicate HFD increased sEPSC frequency in D2 like MSNs, but as above, sEPSC data was not separated by D1 vs. D2 like cells.
-In the discussion (line 259-260) the authors note that AMPH increases PPR (presumably they still mean in D2 like neurons from HFD animals) but results (Fig. 2C) show a decrease in PPR in that group, and an increase in P1 amplitude (Fig. 2D).
-I appreciate that the intent of the authors in text noted in the prior two comments was to consider apparently contradictory results, but they don’t seem to accurately describe results in hand, and I can’t make clear sense of proposed explanation for apparent contradictions.
-Overall, the cell phys data suffers from insufficient rigor in clearly / effectively isolating D1 vs. D2 MSNs, and also from what seem to be pretty minimal clear effects of HFD in any specific MSN population.
Additional comments:
On the other experimental results:
-Increased inflammation associated with HFD is relatively well established and the discussion might be able to be more effective at highlighting novelty of current results vs. existing data.
-Changes in DOPAC absent changes in DA are somewhat surprising / confusing. Maybe changes in MAO expression are involved?
-The results with cyclic voltammetry showing HFD reduces single pulse DA release in control or HFD animals, but increases phasic release across a wide range of frequencies is intriguing, but also confusing. In addition to the citations the authors include (55, 56) there is another recent study (PMID 35900193) that takes a similar approach. Overall, it's not clear this method provides a lot of consistency across studies. Discussion in this manuscript indirectly acknowledges / implies that, but it could do a better job of highlighting most consistent aspects of these studies with the current one, vs. potentially conflicting results.
On the methods:
-Line 390: 10 Hz seems incorrect for single stimuli.
-Line 392: How was Clampfit used to analyze data recorded with an EPC-7? If data were converted to abf, please note how.
-Please specify which method for post-hoc tests in ANOVA was used.
-Line 466: Does this mean current for phasic stim was also 400 uA?
On the writing:
The manuscript has substantial problems with English language and grammar, and extensive detailed revisions would be necessary for publication. The following are just a few examples representative of the types of errors that occur throughout the text:
Line 19: how reward circuit is modified
Line 20: are still being unraveling
Line 40: had overweight or obesity
Line 45: used to combat the obesity
Line 49: exposure to this kind of diets
Line 117:121: refers to Fig. 1 but should be Fig. 2.
Line 126-127: can be identified by exhibit a depression
Line 155: at pre and postsynaptically level
Author Response
REVIEWER Nº2: REPORT FORM
This manuscript by Plaza-Briceno et al. evaluates the effect of a high fat diet on several diverse measures of glutamatergic synaptic signaling, inflammation, and dopaminergic signaling in the nucleus accumbens using a wide range of applicable techniques. The study has a variety of potentially interesting results and does stand out for its diversity of approaches. However, in current form it also suffers from some significant weaknesses as further outlined below.
We appreciate the reviewer's comment.
Major comments:
One the cell physiology (Fig. 2):
1.-Relative to options broadly available, PPR is not a particularly robust way to identify D1 vs. D2 MSNs. At a minimum, it would be reassuring to see PPR correctly predicts sensitivity to a selective D2 agonist in the current animal model.
ANSWER 1
Since we did not want to alter through pharmacology the possible changes induced by diet, we performed our experiments without agonist/antagonist. To better characterize the putative D1 and D2 MSNs, we incorporate a supplementary figure 1 showing the main differences in electrical properties of MSN subtypes described in rat slices, such as membrane resistance, AP threshold, or firing rate used jointly to PPR to classify the recorder cells.
To clarify this point, we have included the following texts:
Highlighted in green; Page 3, lines 117 -120). “…, according to the electrophysiological characterization of MSN described in the dorsal striatum, the D1-MSNs and D2-MSNs can be identified by exhibit a non-facilitating and a facilitation response to paired-pulse protocol, respectively [16, 34-37]”.
Highlighted in yellow; Page 12, lines 397 -401). “Finally, MSN subtypes were classified as D1-like MSNs or D2-like MSNs according to paired-pulse ratio, input resistance, firing properties, and action potential (AP) threshold, being the non-facilitating PPR, lower input resistance, lower firing rate or higher AP threshold characteristics to D1-like MSN, contrary to D2-like cells (see supplementary figure 1)”.
2.-It seems unlikely that the ratio of D1 to D2 MSNs identified by PPR in control vs. HFD animals (10:7 vs. 8:9) is significantly different. This question should be evaluated with appropriate statistical tests and de-emphasized, reinterpreted, or removed if there is no significant difference.
ANSWER 2
We appreciate the reviewer's comment. The graph of the D1-like MSNs and D2-like MSNs identified by electrophysiological criteria previously published includes the cells used in the original graph plus the cells found in animals used in supplementary figure 1. In this context considering the percentages found in control (65.5 % D1-like MSNs and 34.5% D2-like MSNs) and HFD rats (47.1% D1-like MSNs and 52.9% D2-like MSNs), we were able to observe significant differences using a Chi-square test.
To clarify this point, we have included the following texts:
Highlighted in green; Page 3, lines 124 -126). “Interestingly, the proportions of D2-like MSNs in HFD rats (52.9%) were significantly higher than control rats (34.5%) (Fig. 2A; P=0.0077)”.
3.-It’s not entirely clear if the spontaneous EPSC data in Fig. 2A was collected in an entirely different group of cells than the evoked EPSC data in the rest of the figure. It seems it would be possible to collect all of these data in the same set of cells if the sweep length was long enough to collect spontaneous data after the evoked responses. If this is in fact what was done, then it would be interesting to know if spontaneous EPSC frequency differed in presumed D1 vs. D2 MSNs.
ANSWER 3
We appreciate the reviewer's comment. This comment was already answered above for Reviewer 1 (ANSWER 2) “We fully agree with the reviewer's comment. Since most of the sEPSC frequency results in the current Figure 2A were recorded in the same cells as shown in Figure 2B, we have modified figure 2, the figure 2 caption (highlighted in yellow; Page 5, lines 161 -170), and the description of the results (highlighted in yellow; Pages 3 - 4, lines 120 -140)”.
4.-Similarly, are the single pulse experiments in Fig. 2D an analysis of the first of the paired pulses in Fig. 2C, or are results in these panels from separate experiments in a separate group of cells? Either is acceptable, but text should be clear on this point.
ANSWER 4
We appreciate the reviewer's comment and suggestion. We introduce the following changes in the text of the Result section and figure captions:
Page 4, lines 147 - 150
“Furthermore, the EPSC amplitude of control rats did not change in the presence of AMPH either D1-like MSNs (before: 56.47 ± 13.77 and after: 79.03 ± 18.03, n=7, P = 0.109) or D2-like MSNs (before: 52.31 ± 13.53 and after: 67.53 ± 13.92, n=5, P = 0.625)(Fig. 2D, left panel)”.
Page 6, lines 161 - 184
“Figure 2. Changes in the NAcc glutamatergic synaptic transmission of control and HFD male rats. (A) Representative PPR average traces were recorded in D1-like MSNs and D2-like MSNs of the control group (light and dark blue, respectively) and in D1-like MSNs and D2-like MSNs of HFD (light and dark orange, respectively). Summary graphs of PPR average values in control (left) and HFD (middle) groups, showing depression in D1-like MSNs and facilitation in D2-like MSNs in both groups. The proportion of D1-like and D2-like cells in control and HFD groups (right). (B) Representative traces of spontaneous synaptic activity were recorded in D1 (upper traces) and D2 (lower traces) -like MSNs of control (left) and HFD (right) in basal conditions. The lower panel shows the summary graphs of spontaneous EPSC frequency in basal and AMPH conditions (0.1 uM) for two cell types, comparing the mean values between groups. (C) Superimposed PPR average traces were recorded in representative D1-like MSNs (upper) and D2-like MSNs (lower) control cells in basal and in the presence of AMPH (light and dark blue, respectively). Summary graphs of PPR in control (left) in D1-like and D2-like cells, comparing the average value in basal and AMPH conditions. Summary graphs of PPR in HFD (left) in D1-like and D2-like cells, comparing the average value in basal and AMPH conditions. (D) Average EPSC evoked by a single pulse recorded in single D1-like and D2-like cells of control rats, superimposing the traces recorded in basal and AMPH conditions (upper panel). Summary graphs of average values of EPSC amplitude in control in D1-like and D2-like cells, comparing the average value in basal and AMPH conditions (lower panel). Average EPSC evoked by a single pulse recorded in single D1-like and D2-like cells of HFD rats, superimposing the traces recorded in basal and AMPH conditions (upper panel). Summary graphs of average values of EPSC amplitude in D1-like and D2-like cells, comparing the average value in basal and AMPH conditions (lower panel). Data are expressed as means ± SEM (mean standard error), and the statistical analysis used were t-test Welch one-tailed (Fig. 2A), Mann-Whitney two-tailed two-way (Fig.2B), and Wilcoxon two-tailed (Fig. 2C-D) (✱P <0.05)”.
5.-While I believe separating results by D1 vs. D2 is still highly valuable, recent work has begun to shed some doubt on the canonical idea that these receptors are good markers of the direct vs. indirect pathways (e.g. see PMIDs 26214370 and 30622165), and I think the authors should probably consider this in the discussion.
ANSWER 5
We appreciate the reviewer's comment and suggestion. We introduce the following changes in the text to clarify this important topic:
Highlighted in green; Page 9, lines 276 - 283: “Finally, the classical implications associated with the change in the activity of the D1-MSNs and D2-MSNs, or the changes in the expression patterns of D1 and D2 receptors have been related to exclusive effects on direct or indirect pathways in NAcc. However, this topic is being reconsidered, since studies that have combined electrophysiological, optogenetic, and chemogenetic techniques have shown in NAcc that D1-MSNs innervate the ventral midbrain (direct pathway) and the ventral pallidum (indirect pathway), which receives close about 90% and 50% of D2-MSNs and D1-MSN afferents, respectively [19, 49]”.
6.-In the discussion (line 257) the authors state that results indicate HFD increased sEPSC frequency in D2 like MSNs, but as above, sEPSC data was not separated by D1 vs. D2 like cells.
ANSWER 6
We appreciate the reviewer's comment. This comment was also pointed out above for Reviewer 1 (ANSWER 5). Thus, it was already clarified, modifying Figure 2, and introducing these data in the result and discussion sections.
7.-In the discussion (line 259-260) the authors note that AMPH increases PPR (presumably they still mean in D2 like neurons from HFD animals) but results (Fig. 2C) show a decrease in PPR in that group, and an increase in P1 amplitude (Fig. 2D).
ANSWER 7
We appreciate the reviewer's comment. We made a type of mistake, and as shown in figure 2C, in putative D2 cells of the HFD group, the PPR is diminished by AMPH, which corresponds to an increase in the neurotransmitter release probability. Thus, we corrected the sentence:
Highlighted in green; Page 9, lines 261 - 263
“This decrease in sEPSC frequency induced by AMPH occur along with the increase of glutamate release probability (i.e., PPR; Fig. 2C-D) [31, 48]”.
8.-I appreciate that the intent of the authors in text noted in the prior two comments was to consider apparently contradictory results, but they don’t seem to accurately describe results in hand, and I can’t make clear sense of proposed explanation for apparent contradictions.
ANSWER 8
We agree with the comment, and we try to clarify this point rewrite the following sentence:
Highlighted in yellow; Page 9, lines 263 - 269
These contradictory AMPH effects could be explained by indirect activation of D2 receptors that 1) downregulate the firing rate of action potentials (AP) on the glutamatergic terminals (i.e., decreasing the sEPSC frequency) and/or 2) inhibit the lateral inhibition on glutamatergic and GABAergic terminals on MSNs. However, the regulation by hetero-receptors on cortical and dopaminergic inputs in NAcc (e.g., endocannabinoid, acetylcholine, and adenosine, among others) is an open question in our work [33, 49].
9.-Overall, the cell phys data suffers from insufficient rigor in clearly / effectively isolating D1 vs. D2 MSNs, and also from what seem to be pretty minimal clear effects of HFD in any specific MSN population.
ANSWER 9
We agree with the comment. This point was already answered above in ANSWER 1 (to Reviewer 2) and ANSWER 1B (to Reviewer 1).
Additional comments:
On the other experimental results:
-Increased inflammation associated with HFD is relatively well established and the discussion might be able to be more effective at highlighting novelty of current results vs. existing data.
We thank the comment. On Page 10 Lines 304 -313, we try to relate that the neuroinflammation induced in our model of obesity may be part of the pathological mechanism that leads to the functional alterations observed in NAcc. On the other hand, the NLRP3 inflammasome proteins of the NAcc could be a pharmacological target to control the hedonic imbalance in obesity.
-Changes in DOPAC absent changes in DA are somewhat surprising / confusing. Maybe changes in MAO expression are involved?
We appreciate the reviewer's comment. Indeed we were also surprised by the effects of chronic HFD exposure on DA metabolism. We do not rule out that MAO is involved in the reduction of DA metabolism, either by a decrease in the expression or the activity of the enzyme.
-The results with cyclic voltammetry showing HFD reduces single pulse DA release in control or HFD animals, but increases phasic release across a wide range of frequencies is intriguing, but also confusing. In addition to the citations the authors include (55, 56) there is another recent study (PMID 35900193) that takes a similar approach. Overall, it's not clear this method provides a lot of consistency across studies. Discussion in this manuscript indirectly acknowledges / implies that, but it could do a better job of highlighting most consistent aspects of these studies with the current one, vs. potentially conflicting results.
Thank you for this comment. We clarify this point, by rewriting the following sentence:
Highlighted in green; Page 10, lines 313 - 318
“It has recently been shown using FSCV ex-vivo that exposure to HFD from postnatal days 42 to 84 decreases phasic DA release (5 pulses 20 Hz) and reuptake rate in NAcc slices from mice of both sexes [63]. Interestingly, after restoring baseline values of tonic (monophasic) DA release, the perfusion of ACSF plus IL-6 (10 nM) or ACSF plus TNFα (300 nM) significantly reduces NAcc DA release in mice fed with HFD [63]”.
On the methods:
-Line 390: 10 Hz seems incorrect for single stimuli.
Thank you for this comment. We deleted “10 Hz” from the text (Page 12, Line 412).
-Line 392: How was Clampfit used to analyze data recorded with an EPC-7? If data were converted to abf, please note how.
ANSWER 9
Thank you for this comment. We clarify this point, by rewriting the following sentence:
Highlighted in yellow; Page 12, lines 399 - 401
“eEPSCs and sEPSC were recorded and imported as HEKA pulse data by Microcal Origin 6.0 software to be analyzed offline and graphed, using analysis software (Clamfit 6.0, Molecular Devices, USA)”.
-Please specify which method for post-hoc tests in ANOVA was used.
Thank you for this comment. For the analysis 2-way ANOVA, we used Bonferroni post-hoc test (Page 14, Line 497).
-Line 466: Does this mean current for phasic stim was also 400 uA?
Thank you for this comment. Yes, phasic DA release was stimulated using 400 μA (Page 14, Line 487).
On the writing:
The manuscript has substantial problems with English language and grammar, and extensive detailed revisions would be necessary for publication. The following are just a few examples representative of the types of errors that occur throughout the text:
ANSWER 9
Thank you for this comment. We have requested an expert to review the manuscript to correct grammatical errors.
Line 19: how reward circuit is modified: Grammatical error fixed.
Line 20: are still being unraveling: Grammatical error fixed.
Line 40: had overweight or obesity: Grammatical error fixed.
Line 45: used to combat the obesity: Grammatical error fixed.
Line 49: exposure to this kind of diets: Grammatical error fixed.
Line 117:121: refers to Fig. 1 but should be Fig. 2: Grammatical error fixed.
Line 126-127: can be identified by exhibit a depression: Grammatical error fixed.
Line 155: at pre and postsynaptically level: Grammatical error fixed.

Round 2
Reviewer 1 Report
I am satisfied with the author's responses to my questions/issues raised in my initial review. The revised manuscript is easier to follow. In my opinion the revised paper can be accepted.
Author Response
On behalf of the authors, I want to thank you so much for the positive and outstanding review made by reviewer 1. It has been a pleasure to respond to the reviewer's comments.